# Investigation of Alcohol-Drinking Levels in the Swiss Population: Differences in Diet and Associations with Sociodemographic, Lifestyle and Anthropometric Factors

**DOI:** 10.3390/nu14122494

**Published:** 2022-06-16

**Authors:** Dasom Bae, Anna Wróbel, Ivo Kaelin, Giulia Pestoni, Sabine Rohrmann, Janice Sych

**Affiliations:** 1Institute of Food and Beverage Innovation, ZHAW School of Life Sciences and Facility Management, Grüentalstrasse 14, CH-8820 Wädenswil, Switzerland; dsbae616@gmail.com (D.B.); janice.sych@zhaw.ch (J.S.); 2Institute of Computational Life Sciences, ZHAW School of Life Sciences and Facility Management, Grüentalstrasse 14, CH-8820 Wädenswil, Switzerland; anna.wrobel@zhaw.ch (A.W.); ivo.kaelin@zhaw.ch (I.K.); 3Division of Chronic Disease Epidemiology, Epidemiology, Biostatistics and Prevention Institute, University of Zurich, Hirschengraben 84, CH-8001 Zurich, Switzerland; giulia.pestoni@ffhs.ch; 4Nutrition Group, Health Department, Swiss Distance University of Applied Sciences, Zollstrasse 17, CH-8005 Zurich, Switzerland

**Keywords:** alcohol-drinking level, menuCH, 24 h dietary recall, energy intake, macronutrient, Alternate Healthy Eating Index

## Abstract

Alcohol-drinking levels in Switzerland were investigated to identify dietary differences and explore the relationship between drinking levels and sociodemographic, lifestyle and anthropometric factors using the National Nutrition Survey menuCH (*n* = 2057, 18–75 years). After two 24 h dietary recalls (24HDRs), participants were categorized into four subgroups: abstainers (both self-declared alcohol avoidance and no alcohol reported); no alcohol reported; moderate drinkers (women/men < 12 g/<24 g mean daily alcohol, respectively); and heavy drinkers (women/men > 12 g/>24 g mean daily alcohol, respectively). Differences in diet between these groups were described by comparing daily total energy and non-alcohol energy intake, macronutrient energy contribution, food group intake, and diet quality (Alternate Healthy Eating Index excluding alcohol). The sociodemographic, anthropometric and lifestyle factors that determine alcohol-drinking levels were investigated using multinomial logistic regression. Abstainers reported the lowest daily energy intake (total and non-alcohol), heavy drinkers had the highest total energy intake and the lowest diet quality, and moderate drinkers had the highest non-alcohol energy intake. Sex, age, language region, body mass index, household size, smoking status, self-reported health status and following a diet were significantly associated with different alcohol-drinking subgroups. Results could facilitate interventions that target subgroups who exceed safe alcohol-drinking levels and lead unfavorable lifestyles.

## 1. Introduction

Alcohol use is a major risk factor for disease globally. It has been linked with increased risk of communicable diseases, non-communicable diseases (NCDs) and injuries, and attributed to 5.0% of disability-adjusted life years (DALYs) and 5.3% of all deaths [1]. Considering this public health burden, many national health authorities worldwide have implemented safe-drinking guidelines, which were proposed by the World Health Organization in 2001 [2,3]. These are typically given as average intake per day and refer to a standard drink, which is often defined differently across countries. In Switzerland and most EU countries, one standard drink is 10–12 g pure alcohol, and safe-drinking levels are limited to one drink for women and two drinks for men per day [2,3]. Some EU countries have guidelines with higher alcohol thresholds and these might need to be reduced according to recent calculations of lifetime risk of mortality [4]. Alcohol-drinking guidelines in most countries also promote alcohol-free days and include additional restrictions for higher-risk population subgroups such as women during pregnancy. The UK introduced a weekly alcohol-drinking limit of 112 g for both sexes (which corresponds to 14 g of alcohol per day) and added a general warning about the increased risk of specific cancers [5].

Alcohol provides a high amount of energy (7 kcal/g), but has limited or no nutritional value and affects brain function pharmacologically by complex interactions with multiple neurotransmitters [6]. Furthermore, alcohol can trigger appetite by its diverse effects on central and peripheral satiety signals, together with a learned association between alcohol and eating [6,7]. As reviewed in the literature, alcohol has a high thermogenic effect leading to increased energy expenditure, which might partly account for obesity-protective effects observed in moderate-level drinkers [7,8].

Given alcohol’s high energy content and its potential to affect appetite, many studies have investigated the association between alcohol intake and obesity [7,9,10,11]. Some studies have suggested that heavy drinking is positively associated with weight gain compared to light-to-moderate alcohol intake, whereas others reported a negative relationship or no relationship between alcohol intake and body weight [7]. Moderate drinkers were more likely to obtain additional energy from alcoholic beverages without compensating for this additional energy, but this was not associated with weight gain over the long term [6]. In contrast, the risk of obesity was more consistently shown for heavy drinkers, despite reported meal-skipping habits [7,8]. These contradictory results are likely caused by the overall complexity of alcohol metabolism, its multiple effects on the body and individual differences, including sex [6,8].

Evidence suggests that alcohol intake can impact dietary behavior in various ways leading to differences in food and macronutrient intake and overall diet quality. In the European Prospective Investigation into Cancer and Nutrition (EPIC), the diet of participants with high alcohol intake was higher in fat and protein energy, but lower in sugar energy contribution compared to abstainers [12]. In the National Health and Nutrition Examination Survey (NHANES), participants consumed more discretionary oils and solid fat, but less milk on alcohol-drinking days than non-drinking days. In these results, differences in total dairy intake were not observed for men [13]. Moreover, increased intake of alcoholic beverages was associated with reduced diet quality scores [14]. In Switzerland, decreasing trends in alcohol intake have been reported based on taxation and intake frequency data [15,16]. However, studies are lacking on alcohol intake and overall diet, and on determinants of alcohol drinking in the Swiss population.

Using the menuCH survey, our study investigated alcohol-drinking levels in the Swiss population with the aim to describe differences in diet and to explore associations with sociodemographic, lifestyle and anthropometric characteristics. 

## 2. Materials and Methods

### 2.1. Study Design and Setting of MenuCH Survey

A cross-sectional Swiss National Nutrition Survey, menuCH, was conducted between January 2014 and February 2015 in 10 study centers across Switzerland, with details described earlier [17,18]. The survey used a stratified random sample of Swiss residents 18 to 75 years old, intended to be representative of the seven major regions (Lake Geneva Region, Espace Mittelland, Northwestern Switzerland, Zurich, Eastern Switzerland, Central Switzerland and Ticino; Eurostat NUTS-2), covering the three main language regions (German-, French- and Italian-speaking regions) and five age groups (18–29, 30–39, 40–49, 50–64 and 65–75 years). From a gross sample of 13,606 individuals, 5496 were contacted and 2086 agreed to participate, giving a net participation rate of 38%. The survey protocol was approved by the ethical committee of the city of Lausanne (Protocol 26/13) and by the corresponding regional ethical committees, and registered in the International Standard Randomised Controlled Trial Number registry (ISRCTN 16778734). The study was conducted according to the guidelines of the Declaration of Helsinki, including written consent by all participants. Using the data of 2057 menuCH study participants who completed two 24 h dietary recalls (24HDRs), the present study is a secondary analysis which followed the guidelines of STROBE-nut [19].

### 2.2. Assessment of Food and Alcohol Intake 

All foods and beverages consumed by the menuCH participants on two non-consecutive 24HDR interview days were recorded by a trained dietitian; the first interview was in person and the second was by telephone two to six weeks later [17]. Reporting of food and beverage intake was facilitated by a food picture book consisting of 119 series of 6 graduated portion sizes and about 60 household measures. All intakes were recorded using the trilingual Swiss version (0.2014.02.27) of the software GloboDiet^®^ (formerly EPIC-Soft^®^, International Agency for Research on Cancer IARC, Lyon, France) [20], adapted by the Federal Food Safety and Veterinary Office, Bern, Switzerland; an updated version of the software (0.2015.09.28) was used in data cleaning. Study participants also completed a questionnaire (menuCH questionnaire) on dietary habits, and sociodemographic and lifestyle variables. The questionnaire also inquired about the avoidance of certain foods and alcohol, without specifying duration. Only the answers concerning alcohol avoidance were used in our analysis to investigate alcohol abstainers.

For each 24HDR interview, the total intake of all alcoholic beverages was converted to grams of alcohol per day using the Swiss Food Composition Database [21]; then, the mean daily intake per participant was calculated from the two 24HDRs.

Study participants were categorized into four alcohol-drinking subgroups, using the 24HDR alcohol intake data and the answers about alcohol avoidance. Considering non-drinkers, the subgroup of abstainers was defined as study participants who reported no alcoholic beverage intake at both 24HDRs and also reported alcohol avoidance in the menuCH questionnaire, whereas non-drinkers who did not declare alcohol avoidance were assigned as no alcohol reported (NAR). Using the Swiss Federal Office of Public Health (FOPH) guidelines for safe alcohol intake, alcohol-drinking participants were categorized as moderate drinkers when the average daily intake of alcohol was <12 g and <24 g for women and men, respectively, and as heavy drinkers when the alcohol intake was >12 g and >24 g for women and men, respectively [22]. These amounts correspond to one standard drink for women (i.e., 140 mL of wine at 12% alcohol by volume (ABV), 330 mL of beer at 5% ABV, 70 mL of liquor or aperitif at 25% ABV and 40 mL of spirits at 40% ABV) and two standard drinks for men.

### 2.3. Sociodemographic, Lifestyle and Anthropometric Variables

Using the data derived from the menuCH questionnaire and anthropometric measurements, the following variables were considered: sex, language-speaking region, age, body mass index (BMI), nationality, income, education, household size, marital status, physical activity, smoking status, self-reported health status and current diet status [23]. To summarize, age was calculated based on the self-reported birth date and categorized into four groups (18–29, 30–44, 45–59 and 60–75 years) [24]. Language region of participants was defined by the canton of residence (German-speaking: Aargau, Basel-Land, Basel-Stadt, Bern, Lucerne, St. Gallen and Zurich; French-speaking: Geneva, Jura, Neuchatel and Vaud; and Italian-speaking: Ticino), and these were stratified into three language-speaking regions (German-, French- and Italian-speaking regions). Physical activity level was assessed by the short-form International Physical Activity Questionnaire (IPAQ) and categorized as low, moderate or high [25]. Income level and highest level of education attained were also investigated. For income and physical activity, data were missing for 585 and 473 of study participants, respectively (27.6% and 24.2%, weighted data) [24]. Following international standard protocols, BMI was calculated from measurements of body weight and height, then categorized as underweight, normal, overweight and obese (<18.5, 18.5 ≤ 25.0, 25.0 < 30.0 and ≥30 kg/m^2^) [26]. Self-reported data was used when these measurements were not possible (*n* = 7), and for pregnant and lactating women (*n* = 27) [17]. The questionnaire also included self-reported health and a binary question about following a weight-loss diet.

### 2.4. Dietary Outcomes

Food items reported from the 24HDRs were regrouped from 19 categories (GloboDiet^®^) to 13 food groups (excluding alcoholic drinks and recipes), which were then used to assess the mean daily food intake of study participants in the 4 alcohol-drinking subgroups. The selection of the food groups was based on the national food-based dietary guidelines (FBDG) [27] and on the food groupings used in previous studies [13,24]. The investigated food groups were all types of cereals and starchy foods (including potatoes); vegetables; fruits; all dairy products and the subgroups of milk, yoghurt and cheese; all meat and the subgroups of processed meat, red meat, white meat and other meat types; fish and seafood; eggs; and nuts, olives and legumes combined. Non-alcoholic beverages were investigated as soft drinks and all other energy drinks; fruit or vegetable juices; and other beverage types. Discretionary oil and fat were combined; added sweeteners containing sugar (jams, honey and sweet sauces), chocolate products and all sweet foods (cakes, tartes, pies, pastries, cookies and milk-based desserts) were combined on the basis of their higher sugar content compared with other foods. Savory snacks, seasoning and sauces were combined into one group considering their high salt content. All remaining foods such as cream, milk and meat substitutes, and soups were combined as others.

The mean and standard error of the mean (SEM) from the two 24HDR interviews were used to describe the dietary outcomes including total energy intake, non-alcohol energy intake, non-alcohol energy contribution from the macronutrients (carbohydrates, protein and fat) and food group intake. Non-alcohol energy intake (kcal/day) was calculated from the energy of all food and beverages consumed except for ethanol energy.

### 2.5. Diet Quality 

Diet quality can be assessed by indices such as the Healthy Eating Index (HEI), the Mediterranean Diet Score (MDS) and the Healthy Diet Indicator (HDI), which are based on compliance to dietary guidelines [28,29,30], whereas the Alternate Healthy Eating Index (AHEI) has the advantage of reflecting both compliance to dietary guidelines and risk of major chronic diseases [31]. In our study, a modified version of the AHEI 2010 was used to assess diet quality of study participants based on 10 components: vegetables, fruits, whole grains, sugar-sweetened beverages and fruit juices, nuts and legumes, red and processed meat, trans fat, fish, polyunsaturated fatty acids (PUFA) and sodium. The alcohol component was excluded since alcohol was the variable of interest. Each single component score can range from 0 to 10 points, and the total score (sum of the 10 components) can range from 0 to 100 points, indicating low to high diet quality [23,31]. Based on the mean of two 24HDRs, the AHEI was calculated for each participant belonging to an alcohol-drinking subgroup. Study participants were categorized into tertiles based on their AHEI score.

### 2.6. Regression Analysis

Multinomial logistic regression was conducted to predict the odds of study participants belonging to one of the four alcohol-drinking subgroups using the independent variables of sex, age, language region, BMI, nationality, education, physical activity level, household size, marital status, income level, smoking status, self-reported health status and current weight-loss dieting status. The model was also adjusted for season and day of the week. Missing values were imputed by multivariate imputation using chained equations m = 25, mice package in R [32]. Across the drinking subgroups, the NAR group was used as the reference since it was the largest subgroup (678 participants, 32.9%) [33]. Odds ratios and 95% confidence intervals were calculated for each variable level and statistical significance was set at *p* < 0.05.

### 2.7. Statistics and Weighting Strategy

All descriptive and statistical analyses were carried out with R (version 4.0.5), with additional R packages for weighting (stats), multiple imputation (mice) and regression analysis (nnet) [32,34,35].

To correct for sampling design and non-response, the survey weighting factors for sex, age, marital status, the major area of Switzerland, household size and nationality were applied to all data, with additional weighting factors for season and weekday applied to all food and beverage consumption data (including AHEI) [36]. This strategy aimed to more accurately extrapolate the results from 2057 participants to 4,627,878 individuals of the Swiss population.

## 3. Results

Table 1 shows the characteristics of the study participants by alcohol-drinking subgroups, where 58% of the study participants were alcohol drinkers, i.e., moderate or heavy drinkers. Abstainers were the smallest group with 9.3% of all participants, and the NAR group comprised almost one third of the participants, followed by moderate drinkers. The safe-drinking level for alcohol recommended in Switzerland was exceeded by 28% of the population (30% and 25% of men and women, respectively, weighted data), assigned to the subgroup of heavy drinkers. Furthermore, women were overrepresented in the non-drinking groups, in contrast to the higher percentages of men in each drinking group. The older-age subgroups (45–59 and 60–75 years) accounted for a relatively larger portion of heavy drinkers. Study participants who never smoked were mostly abstainers, whereas current smokers represented 34.8% of heavy drinkers.

Abstainers were more represented in the highest AHEI tertile, corresponding to higher diet quality, whereas heavy drinkers were more represented in the lowest tertile (i.e., lower diet quality). Compared with the overall population, study participants belonging to abstainers and the NAR group had higher AHEI scores, whereas moderate drinkers and heavy drinkers had lower AHEI scores.

Some differences in food intake were observed across the alcohol-drinking levels (Table 2). The subgroup of heavy drinkers showed the highest consumption of meat (total meat, processed meat and red meat), cereals and potatoes, fish and seafood, and vegetables, but the lowest consumption of fruit, dairy products, non-alcoholic beverages and sugar-containing foods. Milk and white meat were consumed in the highest amounts by abstainers. Non-alcoholic beverages were consumed in higher amounts by non-drinkers than drinkers.

Table 3 describes the mean daily intake of total energy, non-alcohol energy and alcohol energy, as well as the percentage of non-alcohol energy obtained from macronutrients by alcohol-drinking levels. Data suggest that heavy drinkers had the highest daily intake of total energy and moderate drinkers had the highest intake of non-alcohol energy. Total energy and non-alcohol energy intakes were the lowest among abstainers. The daily carbohydrate energy contribution was slightly higher for abstainers, whereas fat and protein energy were slightly higher for heavy drinkers compared with all other subgroups. Compared to the NAR group, moderate drinkers had a similar energy contribution from carbohydrates and protein, whereas their fat contribution was similar to heavy drinkers.

Table 4 shows the odds ratio of belonging to one of the investigated alcohol-drinking subgroups using NAR as the subgroup reference. Women were less likely to be moderate drinkers than men. Participants from the French-speaking region were more likely to be heavy drinkers compared with the German-speaking region. Participants from the age groups 45–59 and 60–75 were more likely to be abstainers and heavy drinkers than the 30–44 age group. The age group 18–29 was less likely, whereas the group 45–59 was more likely, to be moderate drinkers than the 30–44 age group. The subgroup of abstainers was more likely to be couples with children compared to those without children, have very poor to medium compared to good to very good health status, and have high BMI (obese) compared to normal BMI. Former and current smokers were less likely to be abstainers and more likely to be heavy drinkers compared to never smokers.

## 4. Discussion

Our analysis of the menuCH survey data revealed that 58% of the population were alcohol drinkers and 28% of the population exceeded the upper daily limit for alcohol intake defined by the national health authorities in Switzerland. The odds of belonging to the investigated drinking groups (abstainers, no alcohol reported, moderate and heavy drinkers) differed significantly by sex, language region, age, BMI, household size, marital status, smoking status, self-reported health status and current weight-loss dieting status of study participants. We also observed differences in energy and macronutrient intake as well as diet quality between the alcohol-drinking-level subgroups.

### 4.1. Alcohol-Drinking Levels and Adherence to Guidelines

Our results show that more than a quarter of the population in Switzerland exceeded the safe-drinking limits, and therefore, could be confronted with alcohol-related health risks according to recent evidence [1,4]. This corresponds to 25% of women with daily alcohol intake ≥ 12 g and 30% of men with daily intake ≥ 24 g [22]. In a European study on lifetime risk of mortality, the examination of alcohol consumption data for the German population showed 34.7% and 19.8% of women consumed ≥ 10 g/day and ≥20 g/day, respectively, and 41.5% and 31.7% of men consumed ≥ 20 g/day and ≥30 g/day, respectively [4]. In the same study, the prevalence of the Italian population consuming alcohol at these levels was lower. Results from the French NutriNet Cohort revealed that 21 % and 53 % of women and men, respectively, exceeded 10 g/day of alcohol per day, assessed by three web-based 24HDRs in combination with frequency questions [37].

Comparisons of the prevalence of non-drinkers across studies are limited by differences in definitions, methods of data collection and timeframes considered in the assessment [38]. Our estimation of abstainers was 9.3% based on a survey question about alcohol avoidance that did not query specifically about duration. In comparison, the 2017 Swiss Health Survey (SHS) reported 18% current non-drinkers, with 10% being lifetime non-drinkers and 8% past drinkers [15]. Compared with the age range in the menuCH survey (18 to 75 years), the SHS assessed a wider age group (>15 years) and was based on alcohol consumption frequency. Estimation of lifetime alcohol avoidance by the WHO was 7.9 % and 9.8% in Germany and France, respectively, based on survey questions from national samples > 15 years old [39].

### 4.2. Dietary Differences according to Alcohol-Drinking Levels

Alcohol could contribute to weight gain due to its high energy content and appetite-stimulating effects, which have been well demonstrated in studies [6,40]. Our results show that alcohol drinkers had a slightly higher total energy intake and non-alcohol energy intake than non-drinkers, therefore suggesting that drinkers did not compensate for additional alcohol energy by eating less. Additionally, our findings suggest that moderate drinkers had a higher daily intake of non-alcohol energy than heavy drinkers. Overall, our results seem to be consistent with previous findings that moderate drinkers were more likely to obtain additional energy from alcoholic beverages without reducing their usual food intake, whereas heavy drinkers tended to reduce their daily energy intake [6,8]. The reviewed evidence highlights many inconsistencies across epidemiological studies and a lack of consensus about the effects of added alcohol energy and the risk of weight gain in drinkers [6,7]. Alcohol metabolism is complex, affected by many variables and with large variations between individuals. Consequently, the scope of future research on alcohol-drinking and dietary behavior will need to widen to include genetic, cultural and newly identified lifestyle factors such as sleep and physical activity.

In our analysis, fat and protein energy seemed to be the highest in heavy drinkers and the lowest in abstainers, whereas carbohydrate energy was the highest in abstainers. These differences reflect the trends reported in previous studies, many of which were sex-specific analyses [12,13]. Some of the dietary changes revealed in the heavy-drinking group, such as increased intake of processed meat, red meat, solid fats and salty snacks have been linked with increased risk of NCDs [41]. In contrast, the dietary choices by abstainers and moderate drinkers tend to be healthier foods, such as fruits, dairy products and white meat. In the NutriNet-Santé Cohort, alcohol at ≥10 g/day was associated with higher intake of processed meat and salt, and lower intake of fruit and vegetables, dairy products (only for women) and dietary fiber [37].

Our analysis of the AHEI suggests that diet quality was highest for study participants in the subgroup of abstainers, and lowest in heavy drinkers. A low AHEI score in menuCH heavy drinkers is consistent with Breslow et al. who reported significant decreases in total HEI-2005 scores with increasing alcohol-drinking levels in the US [14]. Their analysis included the alcohol component in the calculation of the HEI scores, whereas we chose to exclude it. Using NHANES data (2003–2008), the factor ‘alcohol- drinking day’ was linked with lower diet quality for men and women, and also with higher non-alcohol energy for men [13].

### 4.3. Associated Factors of Alcohol-Drinking Levels

Our result that women were less likely to be moderate drinkers (reference NAR subgroup) is consistent with other reports that women generally drink less alcohol than men, in Switzerland and elsewhere [12,15,42]. Additionally, participants from the French-speaking region of Switzerland were more likely to be heavy drinkers compared to those from the German-speaking region. The guidelines for safe alcohol intake seem to be less respected in the French-speaking region than in the German and Italian language-speaking regions, also contributing to the lower diet quality observed in the French-speaking region as assessed by the AHEI [17,23].

Age was significantly associated with alcohol-drinking levels for most of the population subgroups, such that older age groups (45–59 and 60–75 years) were more likely to be abstainers as well as heavy drinkers compared to the reference age group of 30–44. Older adults (60–75 years) might avoid alcohol due to poor state of health and overall concern about alcohol-related risks. Compared with younger adults, older adults are more susceptible to the effects of alcohol, mainly due to age-related physiological changes such as slower metabolism and interactions with medication [42]. Driving factors of alcohol intake in the aging population might include changes in mental state, such as depression and loneliness, and social pressures linked with life transitions. In a study of adults aged 65 to 84 years in four European countries, alcohol consumption was higher in men compared with women, and lower in the oldest age group (80–84 years) [43].

Our study showed that the youngest age group (18–29 years) was less likely to be moderate drinkers compared to middle-aged participants (30–44 years), consistent with a previous report in Switzerland based on alcohol consumption frequency [15]. In the Netherlands a lower frequency of alcohol drinking was reported in younger adults (18–35 years), and was especially low in women, compared to the age group of 35–45 [44]. In our study the age group of 45–59 had a higher likelihood to belong to all alcohol-drinking levels (abstainers, moderate and heavy drinkers), which might reflect the wide heterogeneity across this group.

Obese participants were more likely to be abstainers compared to normal weight participants. Additionally, participants who self-reported poor to medium health status were more likely to be abstainers and less likely to be moderate drinkers compared to the participants with good to very good health status. Awareness of obesity-related health risks might be the motivation to avoid alcohol [45,46]. Independent of body size, health was one of the most frequently given reasons for alcohol avoidance by current abstainers [47]. Concerning heavy drinkers, our results did not show an association with obesity. This differs from previous longitudinal studies, which showed that obese male participants consumed more alcoholic beverages than non-obese [7]. Reverse causation might at least partly explain our results such that obese participants concerned with their weight might avoid high-energy foods, including alcohol [48]. Couples with children were more likely to be abstainers than couples without children, which might be linked with a greater perception of responsibility or being a role model by parents. However, in a recent study, family was not frequently given as a reason for abstinence by current abstainers [47]. In our study, we could not determine whether the participants modified their alcohol-drinking behavior after having children or if they were already abstainers.

Lastly, smoking status was a clear determinant of alcohol-drinking levels, such that current and former smokers were positively associated with heavy drinking and less likely to be abstainers compared to never smokers. Many studies have showed a positive association between smoking and drinking, and there is concern about possible synergies between these two behaviors which could lead to increased health risk [37,49,50]. However, strong social pressure to quit smoking might promote the use of alcohol as a compensating effect [51].

### 4.4. Strengths and Limitations

The examination of food group intakes and analysis of the AHEI across subgroups with increasing alcohol-drinking levels revealed several dietary differences of concern. However, cross-sectional studies cannot inform about temporality and the menuCH questionnaire was not developed specifically for an in-depth examination of alcohol intake. However, the survey did include an avoidance question which allowed this study to examine menuCH alcohol abstainers. The self-reported retrospective 24HDR interviews might lead to the under-reporting of alcohol intake [52]. Participation bias was also an important limitation of the study as participants with a higher education and better health were overrepresented in the survey [17]. For study participants who reported alcohol intake for only one 24HDR interview, our calculation of mean intake for both 24HDRs might have overestimated the percentage of moderate drinkers. Likewise, some binge drinkers might have been incorrectly categorized as moderate drinkers. Since alcohol is an occasionally consumed food, the addition of consumption frequency questions to the survey protocol would have widened the possibilities of analysis and conclusions [53]. Additionally, the duration of alcohol avoidance was not queried in the menuCH survey, which prevented us from distinguishing between different types of abstainers. The association between alcohol-drinking levels and diet quality might be affected by the definition of whether abstainers were lifelong abstainers or past drinkers [38,46] and might also depend on the type of index used to assess the diet.

## 5. Conclusions

This study aimed for a comprehensive investigation of alcohol-drinking levels and resulting differences in dietary choices and quality, and associations with sociodemographic, lifestyle and anthropometric factors. Our results suggest differences in dietary intake and confirmed the significance of associations between sociodemographic, lifestyle and anthropometric factors and alcohol-drinking levels. Given that more than a quarter of the Swiss population consume more alcohol than that which is considered safe by national guidelines, a strengthening of intervention policies is urgently needed to reduce alcohol intake in Switzerland. The results of this study can help to strengthen public health measures and to reach subgroups of the Swiss population with unsafe levels of alcohol intake combined with other risk behaviors.

## Figures and Tables

**Table 1 nutrients-14-02494-t001:** Description of study participants categorized by alcohol-drinking levels (*n* = 2057) ^1^.

Characteristics, %	All	Abstainers ^2^	No alcohol Reported ^3^	Moderate Drinkers ^4^	Heavy Drinkers ^5^
**Participants; No. and** (**%**)	2057 (100)	192 (9.3)	678 (32.9)	621 (29.9)	566 (27.9)
**Sex**					
Men	49.8	35.6	43.8	56.9	54.1
Women	50.2	64.4	56.2	43.1	45.9
**Language regions ^6^**					
German	69.2	64.0	71.6	71.9	65.3
French	25.2	27.4	24.0	21.9	29.6
Italian	5.6	8.6	4.4	6.3	5.1
**Age groups**					
18–29 years	18.8	21.2	25.5	13.4	15.8
30–44 years	29.9	28.2	33.6	31.9	23.9
45–59 years	29.8	32.5	24.0	31.6	33.8
60–75 years	21.6	18.0	16.9	23.1	26.5
**BMI categories**					
Underweight	2.4	2.1	2.9	2.4	1.9
Normal	54.1	55.2	57.8	52.4	51.3
Overweight	30.6	19.8	27.9	35.6	32.0
Obese	12.9	22.8	11.4	9.5	14.8
**Nationality**					
Swiss	61.4	53.1	60.2	65.8	61.0
Swiss binationals	13.8	15.0	15.0	13.3	12.4
Non-Swiss	24.8	31.9	24.8	20.9	26.6
**Education categories**					
Primary	4.7	7.3	5.4	3.8	3.9
Secondary	42.6	46.9	44.7	39.6	41.8
Tertiary	52.6	45.8	49.5	56.6	54.3
**Household size**					
Living alone	18.1	15.9	17.9	18.8	18.3
Adult living with parents	7.1	9.2	9.9	4.8	5.7
One-parent family with children	4.4	6.9	4.9	3.1	4.4
Couple without children	31.7	20.1	29.3	32.9	37.0
Couple with children	32.8	41.1	31.4	34.6	29.8
Others ^7^	5.7	6.8	6.1	5.9	4.8
**Marital status**					
Single	31.1	30.9	36.2	29.3	27.1
Married	52.2	52.8	46.9	54.9	55.5
Divorced	12.1	12.0	10.7	12.1	13.7
Other	4.4	4.2	5.7	3.7	3.7
**Income** (**CHF/month**)					
<6000	17.7	21.1	17.6	18.3	16.0
6000–13,000	39.8	34.2	39.7	39.4	42.3
>13,000	14.9	10.3	14.0	16.0	16.3
**Physical activity level**					
Low	15.1	18.4	15.9	14.2	13.8
Moderate	22.7	17.0	21.0	24.7	24.3
High	40.3	37.3	41.5	40.9	39.2
**Smoking status**					
Never	42.9	62.4	47.4	44.8	28.9
Former	33.6	28.7	29.8	36.9	36.3
Current	23.3	8.9	22.1	18.3	34.8
**Self-reported health status**					
Very poor to medium	12.7	22.8	11.9	9.0	14.3
Good to very good	87.1	77.2	87.5	91.0	85.7
**Currently on diet**					
Yes	5.4	9.4	7.1	3.1	4.6
No	94.4	90.6	92.3	96.9	95.4
**AHEI ^8^ tertile; %**					
T1 (9.13–36.11)	33.5	29.9	32.4	32.5	36.8
T2 (36.14–47.46)	33.0	28.2	33.0	33.5	34.2
T3 (47.48–91.05)	33.5	41.9	34.5	34.0	29.0
**AHEI; median**	42.1	44.1	42.5	41.7	41.0

^1^ Percentages weighted for sex, age, marital status, major area of Switzerland, household size and nationality; AHEI tertile and median were also weighted for season and weekday. ^2^ Self-reported alcohol avoidance and no alcohol reported in both 24HDRs; ^3^ no alcohol reported in both 24HDRs; ^4,5^ based on daily mean alcohol intake from both 24HDRs: ^4^ >0 and <12 g for women, >0 and <24 g for men; ^5^ >12 g for women and >24 g for men. ^6^ German-speaking regions: Aargau, Basel-Land, Basel-Stadt, Bern, Lucerne, St. Gallen and Zurich; French-speaking regions: Geneva, Jura, Neuchatel and Vaud; and Italian-speaking region: Ticino. ^7^ Participants living in a shared flat. ^8^ Without the component of alcohol. Abbreviations: BMI, body mass index; CHF, Swiss francs; AHEI, Alternate Healthy Eating Index; T, tertile.

**Table 2 nutrients-14-02494-t002:** Mean daily intake of selected food groups by alcohol-drinking levels (*n* = 2057, g/day, weighted ^1^).

Food Groups	Total Participants (*n* = 2057)		Non-Drinkers				Drinkers			
			Abstainers ^2^ (*n* = 192)		No Alcohol Reported ^3^ (*n* = 678)		Moderate ^4^ (*n* = 621)		Heavy ^5^ (*n* = 566)	
	Mean	SEM	Mean	SEM	Mean	SEM	Mean	SEM	Mean	SEM
**Cereals and potatoes**	287.2	3.2	272.4	10.5	278.1	5.8	290.0	5.5	299.1	6.2
**Total vegetables**	173.4	2.7	160.6	10.0	164.8	4.7	174.3	4.5	185.9	5.2
**Total fruits**	171.0	3.8	175.9	13.1	172.1	6.5	182.5	6.9	156.8	7.0
**Dairy products**	216.5	4.1	227.6	13.7	232.2	7.8	223.6	7.6	189.1	7.0
Milk	113.4	3.4	135.2	12.3	128.4	6.5	114.9	6.3	88.8	5.5
Yogurt and cheese	103.0	2.1	92.3	7.0	103.8	3.7	108.7	4.0	100.3	3.8
**Meat**	108.9	2.0	98.9	6.8	100.9	3.2	106.8	3.5	122.7	3.9
Processed meat	42.7	1.2	33.3	3.2	40.7	2.1	43.8	2.2	46.8	2.2
Red meat	36.9	1.2	25.7	3.8	29.0	1.8	38.7	2.2	47.0	2.7
White meat	26.8	1.1	38.7	4.7	27.1	2.0	22.2	1.7	27.1	2.0
Others	2.5	0.3	1.2	0.5	4.2	0.7	2.1	0.5	1.8	0.6
**Fish and seafood**	21.0	0.9	22.6	2.8	20.0	1.6	17.2	1.4	25.3	1.9
**Eggs**	13.0	0.5	11.1	1.6	14.1	1.0	12.3	0.8	13.1	0.8
**Nuts, olives and legumes**	12.3	0.6	13.4	1.6	14.5	1.5	10.5	0.9	11.4	1.0
**Non-alcoholic beverages**	299.1	9.4	336.2	39.3	355.5	18.6	288.8	14.8	237.8	14.7
Soft drinks, energy drinks	213.3	8.7	261.2	38.7	265.1	17.4	198.8	13.9	157.1	12.7
Fruit or vegetable juices	79.6	2.9	70.4	8.2	85.9	5.5	82.8	5.3	73.0	5.4
Others	6.2	1.0	4.6	2.6	4.4	1.6	7.1	1.7	7.7	2.3
**Discretionary oil and solid fat**	19.9	0.4	19.7	1.3	18.6	0.6	20.6	0.6	20.7	0.7
**Sugar, chocolate, sweets and desserts**	87.5	1.7	94.2	5.4	84.4	2.7	94.4	3.1	81.7	3.2
**Salty snacks, seasoning and sauces**	54.1	1.2	42.8	3.2	56.2	2.2	54.5	2.2	55.4	2.1
**Others ^6^**	70.6	2.5	99.2	11.0	70.2	4.3	68.0	4.3	63.8	4.5

^1^ Weighted for sex, age, marital status, major area of Switzerland, household size, nationality, season and weekday. ^2^ Self-reported alcohol avoidance (4 missing data) and no alcohol reported in both 24HDRs; ^3^ no alcohol reported in both 24HDRs. ^4,5^ Based on daily mean alcohol intake per participant from two 24HDRs: ^4^ >0 and <12 g for women, >0 and <24 g for men; ^5^ >12 g for women and >24 g for men. ^6^ Included cream, milk substitutes, soups and meat substitutes. Abbreviations: SEM, standard error of the mean.

**Table 3 nutrients-14-02494-t003:** Mean daily total energy, non-alcohol energy, alcohol energy and non-alcohol energy percentage obtained from macronutrients by alcohol-drinking levels (*n* = 2057 ^1^).

	Total (*n* = 2057)		Abstainers ^2^ (*n* = 192)		No Alcohol Reported ^3^ (*n* = 678)		Moderate Drinkers ^4^ (*n* = 621)		Heavy Drinkers ^5^ (*n* = 566)	
	Mean	SEM	Mean	SEM	Mean	SEM	Mean	SEM	Mean	SEM
**Total energy (kcal/day)**	2225.7	16.6	2040.3	52.4	2095.0	28.4	2242.8	28.5	2408.7	32.5
**Non-alcohol energy (kcal/day)**	2129.5	16.0	2039.8	52.3	2094.7	28.4	2174.9	28.2	2150.7	30.6
**Alcohol energy (kcal/day)**	96.3	3.0	0.5	0.2	0.4	0.1	67.8	1.7	258.0	6.2
**Carbohydrates (%)**	44.0	0.2	46.8	0.7	44.7	0.3	44.0	0.3	42.2	0.3
**Fat (%)**	37.7	0.2	35.4	0.6	37.1	0.3	38.1	0.3	38.7	0.3
**Protein (%)**	16.3	0.1	15.7	0.4	16.0	0.2	15.9	0.2	17.1	0.2

^1^ Weighted for sex, age, marital status, major area of Switzerland, household size, nationality, season and weekday. ^2^ Self-reported alcohol avoidance (4 missing data) and no alcohol reported in both 24HDRs; ^3^ no alcohol reported in both 24HDRs; ^4,5^ based on daily mean alcohol intake per participant from two 24HDRs: ^4^ >0 and <12 g for women, >0 and <24 g for men; ^5^ >12 g for women and >24 g for men. Abbreviations: SEM, standard error of the mean.

**Table 4 nutrients-14-02494-t004:** Associations between alcohol-drinking levels and sociodemographic, lifestyle, and anthropometric factors (*n* = 2057 ^1^).

NAR ^2^ Subgroup as Reference (*n* = 678)	Abstainers ^3^ (*n* = 192)			Moderate Drinkers ^4^ (*n* = 621)			Heavy Drinkers ^5^ (*n* = 566)		
Characteristics	Participants (*n*)	OR	95% CI	Participants (*n*)	OR	95% CI	Participants (*n*)	OR	95% CI
**Sex**									
Men	57	1	ref.	331	1	ref.	282	1	ref.
Women	135	1.33	[0.92; 1.93]	290	**0.60**	**[0.47; 0.77]**	284	0.82	[0.63; 1.06]
**Language regions ^6^**									
German	112	1	ref.	422	1	ref.	341	1	ref.
French	53	1.28	[0.85; 1.92]	132	1.03	[0.77; 1.36]	166	**1.63**	**[1.23; 2.16]**
Italian	27	1.69	[0.85; 3.37]	67	1.66	[0.99; 2.80]	59	1.32	[0.75; 2.32]
**Age**									
18–29 years	40	0.96	[0.51; 1.79]	90	**0.49**	**[0.32; 0.74]**	87	1.09	[0.71; 1.68]
30–44 years	50	1	ref.	164	1	ref.	113	1	ref.
45–59 years	62	**1.99**	**[1.25; 3.15]**	205	**1.45**	**[1.06; 1.99]**	190	**2.56**	**[1.82; 3.58]**
60–75 years	40	**2.06**	**[1.10; 3.84]**	162	1.46	[0.98; 2.18]	176	**2.92**	**[1.91; 4.46]**
**BMI categories**									
Underweight	5	0.82	[0.26; 2.57]	16	1.01	[0.49; 2.07]	12	0.80	[0.36; 1.80]
Normal	108	1	ref.	322	1	ref.	290	1	ref.
Overweight	42	0.71	[0.46; 1.12]	217	1.14	[0.87; 1.49]	179	1.06	[0.80; 1.41]
Obese	37	**1.77**	**[1.07; 2.93]**	66	0.85	[0.56; 1.27]	85	1.25	[0.85; 1.86]
**Nationality**									
Swiss only	126	1	ref.	464	1	ref.	264	1	ref.
Swiss binational	32	1.14	[0.69; 1.87]	85	0.94	[0.66; 1.32]	79	0.92	[0.64; 1.32]
Non-Swiss	34	1.46	[0.96; 2.22]	72	0.75	[0.55; 1.00]	73	1.20	[0.89; 1.62]
**Education category**									
Primary	15	1.18	[0.57; 2.48]	24	0.84	[0.47; 1.51]	18	0.56	[0.30; 1.02]
Secondary	93	1	ref.	279	1	ref.	264	1	ref.
Tertiary	84	1.05	[0.72; 1.53]	318	1.23	[0.96; 1.58]	284	1.29	[0.99; 1.68]
**Household size**									
Living alone	34	0.99	[0.49; 1.99]	96	1.10	[0.71; 1.69]	95	1.10	[0.70; 1.74]
Adult living with parents	16	2.01	[0.93; 4.34]	33	0.82	[0.47; 1.44]	35	0.86	[0.49; 1.52]
One-parent family with children	15	2.29	[0.94; 5.62]	22	0.78	[0.40; 1.56]	23	1.05	[0.53; 2.05]
Couple without children	48	1	ref.	218	1	ref.	221	1	ref.
Couple with children	69	**2.46**	**[1.49; 4.08]**	215	1.11	[0.80; 1.53]	167	0.91	[0.65; 1.27]
Others ^7^	10	1.65	[0.72; 3.81]	37	1.40	[0.79; 2.46]	25	1.09	[0.60; 2.00]
**Marital status**									
Single	66	1	ref.	171	1	ref.	146	1	ref.
Married	94	0.78	[0.42; 1.43]	362	0.86	[0.58; 1.27]	333	1.21	[0.80; 1.81]
Divorced	25	0.68	[0.33; 1.39]	68	0.83	[0.52; 1.34]	67	0.98	[0.60; 1.62]
Others	7	0.55	[0.22; 1.39]	20	**0.42**	**[0.22; 0.78]**	20	0.54	[0.28; 1.03]
**Income** (**CHF/month**)									
<6000	40	1.5	[0.86; 2.61]	109	1.27	[0.89; 1.81]	90	0.80	[0.54; 1.19]
6000–13,000	68	1	ref.	254	1	ref.	240	1	ref.
>13,000	18	0.84	[0.45; 1.58]	91	1.18	[0.82; 1.71]	83	1.19	[0.80; 1.76]
**Physical activity level**									
Low	29	1	ref.	83	1	ref.	79	1	ref.
Moderate	38	0.86	[0.48; 1.53]	148	1.37	[0.94; 2.00]	131	1.32	[0.86; 2.01]
High	78	0.91	[0.54; 1.52]	257	1.15	[0.81; 1.62]	214	1.14	[0.78; 1.66]
**Smoking status**									
Never	121	1	ref.	282	1	ref.	179	1	ref.
Former	53	**0.64**	**[0.43; 0.94]**	226	1.22	[0.94; 1.59]	207	**1.75**	**[1.31; 2.32]**
Current	18	**0.29**	**[0.16; 0.51]**	113	0.90	[0.65; 1.23]	180	**3.38**	**[2.46; 4.62]**
**Self-reported health status**									
Very poor to medium	48	**2.08**	**[1.30; 3.33]**	62	**0.65**	**[0.44; 0.96]**	83	1.01	[0.70; 1.47]
Good to very good	144	1	ref.	559	1	ref.	483	1	ref.
**Currently on a diet**									
Yes	17	1.19	[0.63; 2.22]	20	**0.49**	**[0.28; 0.85]**	25	0.67	[0.39; 1.14]
No	175	1	ref.	601	1	ref.	541	1	ref.

^1^ Results of multinomial logistic regression adjusted for all variables shown in the table, and for season and weekday; weighted for sex, age, marital status, major area, household size and nationality. ^2^ NAR group (no alcohol reported in both 24HDRs) was the reference for drinking level subgroups. ^3^ Self-reported alcohol avoidance (4 missing data) and no alcohol reported in both 24HDRs; ^4,5^ based on daily mean alcohol intake per participant from two 24HDRs: ^4^ >0 and <12 g for women and >0 and <24 g for men; ^5^ >12 g for women and >24 g for men. ^6^ German-speaking regions: Aargau, Basel-Land, Basel-Stadt, Bern, Lucerne, St. Gallen and Zurich; French-speaking regions: Geneva, Jura, Neuchatel and Vaud; and Italian-speaking region: Ticino. ^7^ Study participants living in a shared flat. ORs in bold indicate significance at *p* < 0.05. Abbreviations: OR, odds ratio; CI, confidence interval; BMI, body mass index; CHF, Swiss francs.

## Data Availability

The entire menuCH dataset and relevant documents (questionnaires, weighting strategy) are available upon request in the data repository: menuch.unisante.ch (access date 25 April 2022).

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
