# Peer review of "Investigation of Alcohol-Drinking Levels in the Swiss Population: Differences in Diet and Associations with Sociodemographic, Lifestyle and Anthropometric Factors"

_nutrients, 2022, doi:10.3390/nu14122494_

Round 1

Reviewer 1 Report

This is a carefully designed, well executed and very nicely written population study that examined the relationships between four levels of alcohol consumption,  highly sophisticated measures of consumed diet as well as major biometric and sociodemographic indices. The sample of  2057 was selected to represent the country of Switzerland. The four groups were: non-drinkers who deliberately abstained from drinking alcohol, non-drinkers on the days they were examined, moderate (or normal) drinkers as well as heavy drinkers.  Swiss standards were used to distinguish moderate from heavy drinkers. Several very nice statistical controls were employed with many of the analyses. Both age and sex differentiated the 4 groups in ways that the authors believe might result in more effective targeted interventions. 

My only recommendation is that the authors specifically provide, in the Abstract, a brief definition of the 4 groups that served as their independent variable.

Author Response

This is a carefully designed, well executed and very nicely written population study that examined the relationships between four levels of alcohol consumption,  highly sophisticated measures of consumed diet as well as major biometric and sociodemographic indices. The sample of  2057 was selected to represent the country of Switzerland. The four groups were: non-drinkers who deliberately abstained from drinking alcohol, non-drinkers on the days they were examined, moderate (or normal) drinkers as well as heavy drinkers.  Swiss standards were used to distinguish moderate from heavy drinkers. Several very nice statistical controls were employed with many of the analyses. Both age and sex differentiated the 4 groups in ways that the authors believe might result in more effective targeted interventions. 

R1.1 My only recommendation is that the authors specifically provide, in the Abstract, a brief definition of the 4 groups that served as their independent variable.

Thank you very much for your comments. Following your suggestion, we have added the definition of the four study groups to the Abstract, shown by Track Changes in the corrected manuscript. We also revised the Abstract to improve its clarify and to stay within the word limit of the Journal.

Reviewer 2 Report

In this manuscript, authors investigate the alcohol-drinking levels in association with diet, and lifestyle choices in Swiss population. The introduction is well laid out and the methods are described well. They have discussed each association in detail in the results section and the associated reasoning is well described in discussion. The strengths and limitations are detailed too. In all, authors have done a good job of laying the information that will be helpful in understanding the relation between alcohol-drinking and lifestyle choices of Swiss population. This is important to understand before any interventions can be made to reduce the alcohol-drinking among the men and women to safe levels.

Minor revisions:

Line 225: The text in the footnote of Table 1 is small compared to rest of the paragraph.

Line 269-270: Space required between the footnote for table 3 and the continued text.

Line 304: Rewrite the sentence for clarity- “In Itlay………in Germany and in our study”.

Line 315: Rewrite for clarity- “Compared with the age………consumption alcohol consumption frequency”.

Author Response

Review 2:

R2.1

In this manuscript, authors investigate the alcohol-drinking levels in association with diet, and lifestyle choices in Swiss population. The introduction is well laid out and the methods are described well. They have discussed each association in detail in the results section and the associated reasoning is well described in discussion. The strengths and limitations are detailed too. In all, authors have done a good job of laying the information that will be helpful in understanding the relation between alcohol-drinking and lifestyle choices of Swiss population. This is important to understand before any interventions can be made to reduce the alcohol-drinking among the men and women to safe levels.

1 Line 225: The text in the footnote of Table 1 is small compared to rest of the paragraph.

  1. Line 269-270: Space required between the footnote for table 3 and the continued text.

Thank you for pointing out those formatting problems: We have corrected the text font in the text underneath Table 1, and also re-checked this point in all the Tables. Additionally space was added after each Table, before the continued text of the manuscript.

  1. Line 304: Rewrite the sentence for clarity- “In Italy………in Germany and in our study”.

Thank you for pointing out problem. We revised this section to improve the clarity by adding some essential information from the study by Shield et al (Addiction 2017, 112, 1535–1544, doi:10.1111/add.13827), lines 313-318.

In a European study on life-time risk of mortality, the examination of alcohol consumption data for the German population showed 34.7 % and 19.8% of women consumed ≥ 10 g/day and ≥ 20 g/day, respectively; and 41.5 % and 31.7 % of men consumed ≥ 20 g/day and ≥ 30 g/day, respectively [4]. In the same study, the prevalence of the Italian population consuming alcohol at these levels was lower.

  1. Line 315: Rewrite for clarity- “Compared with the age………consumption alcohol consumption frequency”.

Following the suggestion of the reviewer, we removed the extra word (consumption) in this sentence, which now reads as follows, lines 328f.

Compared with the age range in the menuCH survey (18 to 75 years), the SHS assessed a wider age group (> 15 years) and was based on alcohol consumption frequency.